# Modeling of Resistive Forces and Buckling Behavior in Variable Recruitment Fluidic Artificial Muscle Bundles

Jeong Yong Kim , Nicholas Mazzoleni and Matthew Bryant *

Department of Mechanical and Aerospace Engineering, North Carolina State University, Raleigh, NC 27695, USA; jkim84@ncsu.edu (J.Y.K.); nwmazzol@ncsu.edu (N.M.)
* Correspondence: mbryant@ncsu.edu

**Abstract:** Fluidic artificial muscles (FAMs), also known as McKibben actuators, are a class of fiber-reinforced soft actuators that can be pneumatically or hydraulically pressurized to produce muscle-like contraction and force generation. When multiple FAMs are bundled together in parallel and selectively pressurized, they can act as a multi-chambered actuator with bioinspired variable recruitment capability. The variable recruitment bundle consists of motor units (MUs)—groups of one of more FAMs—that are independently pressurized depending on the force demand, similar to how groups of muscle fibers are sequentially recruited in biological muscles. As the active FAMs contract, the inactive/low-pressure units are compressed, causing them to buckle outward, which increases the spatial envelope of the actuator. Additionally, a FAM compressed past its individual free strain applies a force that opposes the overall force output of active FAMs. In this paper, we propose a model to quantify this *resistive force* observed in inactive and low-pressure FAMs and study its implications on the performance of a variable recruitment bundle. The resistive force behavior is divided into post-buckling and post-collapse regions and a piecewise model is devised. An empirically-based correction method is proposed to improve the model to fit experimental data. Analysis of a bundle with resistive effects reveals a phenomenon, unique to variable recruitment bundles, defined as *free strain gradient reversal*.

**Keywords:** fluidic artificial muscles; McKibben actuators; variable recruitment



## 1. Introduction

The McKibben actuator has received growing interest in the soft robotics community due to its high power density, inherent compliance, and simple design. It consists of an inner elastic bladder that is wrapped around by an outer braided mesh with a double-helical pattern. One end of the bladder is closed off while the other end is used to provide pressure to the bladder, either pneumatically or hydraulically. As the bladder expands due to internal pressure, the braided mesh constrains the bladder to expand radially while contracting axially. Due to this kinematic relationship of the braided mesh, the McKibben actuator has conventionally been used as a tensile actuator. When a constant pressure is applied, the actuator maintains an equilibrium force–strain state. In terms of control, the actuator can be said to be stable in open loop in response to constant pressure, while any displacement from its equilibrium state results in an opposing force [1]. Thus, it can be characterized as an active spring, a classification that can also be applied to a biological muscle, which is why it is also commonly referred to as a fluidic artificial muscle (FAM), a term that will be used interchangeably in this paper.

Due to this biomimetic behavior of FAMs, researchers have used it in a myriad of robotic applications, including robotic arms, prosthetics, and orthoses [2–7]. Although typically used as contractile actuators, modifications to the design have allowed them to be used as extensile or bending actuators [8]. Recent studies have extended the use of the FAM to multi-chambered bundles with variable recruitment functionality [9–12]. Rather than a single FAM acting in isolation, multiple FAMs are bundled together in parallel connected

by rigid end plates to form a single actuator, as shown in Figure 1. The idea of variable recruitment is biomimetic, based on the hierarchical scheme in which subgroups of muscle fibers, or motor units (MUs), within a mammalian muscle tissue are sequentially activated in order from smallest to largest, a concept known as Henneman's size principle [13]. Compared to a single FAM with equivalent cross-sectional area, the variable recruitment bundle can achieve higher efficiencies. By adaptively recruiting the number of active MUs to meet the load demand, the bundle has the potential to reduce the losses that arise from throttling down the supply pressure. As a result, the variable recruitment bundle has a higher efficiency over a larger force–strain space than a single actuator with equivalent cross-sectional area [9,10]. In addition, by selecting the size and number of FAMs in a MU for a desired force, the force sensitivity can be controlled, allowing for more precise control of the actuator force. The efficiency gains of variable recruitment have been demonstrated experimentally for both pneumatic and hydraulic systems [12,14]. In addition, real-time switching control schemes for variable recruitment have been developed and tested [10,15], and it has been shown that the use of variable recruitment in a hydraulic system with an intermittent pump operation scheme can increase both efficiency and bandwidth when compared to a single actuator [15]. Due to both the compliant and hierarchical nature of variable recruitment, FAM bundles can experience a unique phenomenon during actuation: the compression and buckling of inactive MUs. These inactive FAMs generate a *resistive force* as the bundle contracts, resulting in a bundle force output and free strain that is lower than what would be predicted using a model or experimental characterization of a single FAM and applying it to a bundle [9].

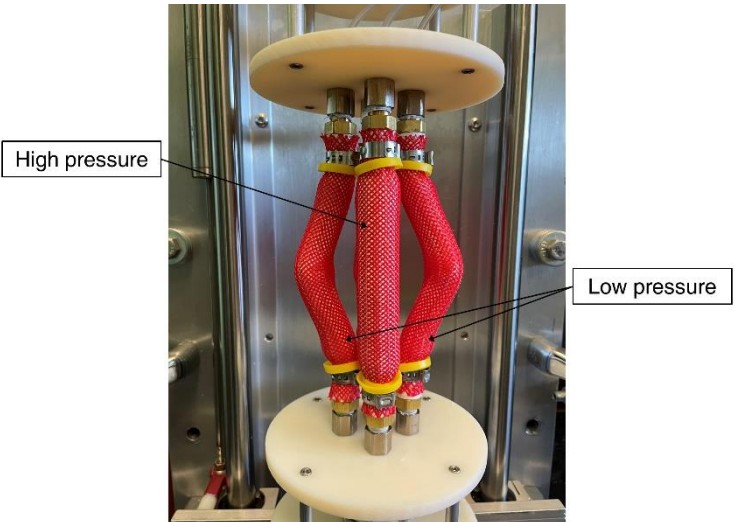

**Figure 1.** Variable recruitment fluidic artificial muscle (FAM) bundle that consists of multiple motor units (MUs) in parallel connected at each end by rigid plates. A MU is the smallest unit of activation and can consist of one or more FAMs. Inactive/low-pressure MUs buckle outward as active MUs contract.

Previous analytical variable recruitment studies have not accounted for resistive forces within a bundle, relying on either ideal virtual-work based models, such as the one presented by Tondu, or semi-empirical corrected models, such as the one presented by Meller et al. [1,14]. These semi-empirical models correct the ideal model by accounting for pressure-dependent free strain that exists due to bladder elasticity, but they do not examine FAM forces past free strain, and therefore cannot account for resistive forces of buckled or post-free strain FAMs within a variable recruitment bundle. In all the variable recruitment simulations or controller developments that exist in the literature, the force exerted by FAMs past free strain is assumed to be zero; in essence, the FAM is assumed to neither contribute positively nor negatively to overall bundle force production. However,

resistive forces have been experimentally observed to contribute negatively to bundle force production [16,17]. In this paper, we propose a method of modeling these resistive forces and study their implications on the overall force output of a variable recruitment bundle. It is important to understand to what extent these resistive forces affect overall bundle performances, as this information will aid in the development of optimal bundle designs for a given application as well as improved variable recruitment controller performance. It should be noted that, although the motivation for modeling the resistive force comes within the context of a variable recruitment bundle, the analysis can be extended to any FAM that is compressed past its free strain.

The remainder of this paper is organized in the following way. In Section 2, conventional methods of modeling the quasi-static force output up to free strain are introduced, giving us insight into how the total force can be decomposed. Section 3 begins with a qualitative description of the different regions of resistive force in FAMs compressed past free strain. The modeling of resistive force for each region is discussed in detail in the same section. In Section 4, the implications of resistive forces on the overall performance of a variable recruitment bundle are explored. In Section 5, an empirically-based correction method is proposed to better match experimental results. Section 6 presents a phenomenon we refer to as bundle *free strain gradient reversal* and discusses its implication to bundle design. The conclusions of the paper are presented in the final section.

## 2. Quasi-Static Modeling of Tensile Force Generation

Prior to modeling the resistive force, we first need an understanding of the quasi-static forces in the tensile force generation regime (i.e., for strains less than the FAM free strain). Chou and Hannaford [18] proposed a model based on the balance between the virtual work done by the equilibrium force $F$ and internal pressure $P$, which is expressed as:

$$- F\delta l = P\delta V ,\tag{1}$$

where $\delta l$ is the variation of axial length and $\delta V$ is the variation of the fluid volume upon which the pressure acts. A kinematic relationship can be derived between the instantaneous radius $r$, and length $l$, of the actuator based on the initial braid angle $\alpha_0$ of the braided mesh, given as:

$$r = r_0 \left( \frac{\sqrt{1 - cos^2\alpha_0 (l/l_0)^2}}{sin\alpha_0} \right) ,\tag{2}$$

where $r_0$ is the initial radius and $l_0$ is the initial length of the braided mesh. By combining these equations, the expression for $F_{mesh}$ can be expressed in terms of the internal pressure $P$, and strain $\varepsilon$.

$$F_{mesh} = \pi r_0^2 P \left( \frac{1}{tan^2\alpha_0} (\varepsilon - 1)^2 - \frac{1}{sin^2\alpha_0} \right)\tag{3}$$

This force will be referred to in this paper as the mesh force, as it expresses the force due to the internal pressure that is converted into axial force using the kinematic constraints of the braided mesh. From this formulation, we can see that the force output is a function of strain. Since the advent of this model, researchers have further developed models to account for the wall thickness of the bladder, tapered geometry of the bladder at its ends, and friction between the bladder and braided mesh [1,19]. Although this model gives us the relationship between the pressure and the axial force output of the actuator due to the kinematic constraint imposed by the braided mesh, it is not able to predict the pressure-dependent nature of free strain. For a given pressure, maximum actuator force occurs at zero strain, which is also known as the blocked force condition. The strain at which the force is zero is known as the free strain, which is constant for all pressures when using Equation (3). Experimental observation of isobaric force–strain curves tells us that the free strain increases with increasing pressure [14]. This pressure-dependent nature of free strain can be accounted for by the bladder elastic forces that oppose the mesh force output. Therefore, the bladder elastic forces are an integral component in understanding the force–strain relationship and have been incorporated into many models. Klute and Hannaford

use the Mooney-Rivlin strain energy function $W$, and apply the principle of virtual work to model the elastic bladder force [20]. Kothera et al. further improved the elastic force term of the model by accounting for changes in bladder thickness [21]. Including the elastic bladder force term, the total force of a FAM is expressed as:

$$F_{total} = F_{mesh} + F_{bladder} = P\frac{dV}{dL} - V_b\frac{dW}{dL} \, ,$$

(4)

where $V_b$ denotes the volume of the bladder. Other models take a force balance approach to model the elastic forces by analyzing the stresses developed in the bladder in the hoop and axial directions [22,23]. The mesh force derived from this approach matches the results from the virtual work balance method, as expected. For this study, the model developed by Klute and Hannaford will be used for forces up to free strain, with the addition of an empirical correction factor that will be discussed in detail in Section 5.

For the tensile force generation regime of the FAM, we note from the models discussed previously that the total axial force production of the actuator can be divided into two components: The contractile force generated by fluid pressure acting on the braided mesh due to the mesh kinematics, and the opposing force that acts against contraction due to the elasticity of the bladder. This idea serves as a basis for understanding the force–strain behavior of a FAM and is valid for force past free strain as well. Therefore, in modeling the resistive forces, the mesh and bladder forces will be modeled separately and summed to give the total force in the resistive force regime.

## 3. Resistive Force Modeling

### 3.1. Experimental Observations of Post-Free Strain FAM Behavior

Prior to presenting the modeling of the resistive force in the post-free strain (compressive) regime, we first present a qualitative description of the post-free strain behavior of a FAM based on our experimental observations. From these observations, the FAM behavior can be divided into two distinct regions: (1) The post-buckling region and (2) the post-collapse region. Immediately after the actuator is compressed past free strain and enters the resistive force–strain regime, the actuator exhibits deflection in the transverse direction due to buckling of the bladder. This behavior is referred to as the post-buckling region. While classical column theory predicts that an axially-loaded, hollow cylindrical column, such as the FAM bladder, will experience linear axisymmetric deformation until a critical load is reached, experimental investigations have established that the classical critical load often overpredicts the load at which buckling occurs in practice [24]. The actual buckling load has been shown to be highly sensitive to geometric imperfections, both in the buckling specimen and in the end constraints of the column [25]. McKibben actuators in practical applications often operate in conditions in which the end constraints may not be perfectly aligned or are subject to perturbations. Therefore, in keeping with our experimental observations, we assume that the FAM bladder enters post-buckling immediately beyond the free-strain condition.

As compressive strain is further increased, a second region of behavior begins when the bladder tube collapses; this region is referred to as the post-collapse region. In the literature on the bending of a hollow cylindrical beam with internal pressure, collapse is considered a structural failure that occurs due to significant deformation of the cross-sectional area of the beam from its circular shape [26]. For a column with clamped-clamped end constraints, the maximum internal moments occur at each end and at the middle. When this moment exceeds the collapse moment, the bladder folds upon itself, effectively acting as a "hinge". The end of the post-buckling region is defined by the collapse moment and the system then transitions to the post-collapse region.

The post-buckling and post-collapse regions make up the post-free strain regime of a McKibben actuator and govern the generation of resistive force. Figure 2 shows the FAM (a) at free length; (b) at free strain, which defines the beginning of the post-buckling region; (c) during the post-buckling region; and (d) during the post-collapse region in

which the internal moment generated has reached the collapse moment. The post-collapse region follows immediately after collapse and continues for the remaining strain range until the maximum strain of the actuator is reached. The following sections will address the quasi-static force modeling of each region.

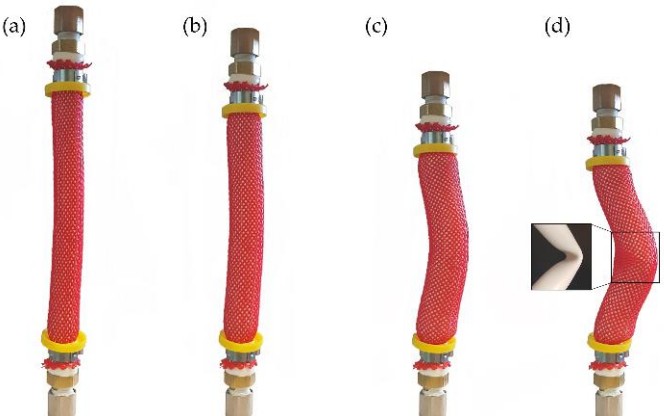

**Figure 2.** The progression of a fluidic artificial muscle (FAM) contracting from its free length (**a**), up to free strain (**b**), and into the resistive force regime. In the post-buckling region (**c**), the bladder is deformed into the first mode shape of a clamped-clamped column. The bladder enters the post-collapse region (**d**) as the bladder cross-section deforms from its initial circular shape.

### 3.2. Post-Buckling Region

The axial force required to maintain static equilibrium of the bladder in the post-buckling shape is solved using the principle of virtual work. For equilibrium, the first variation of the total potential energy, which is the potential energy stored in the bladder due to bending minus the work done to the system by external forces, needs to be zero and is expressed as:

$$F_b \delta x = \delta U_b \,, \tag{5}$$

where $F_b$ is the additional force exerted by the bladder past free strain, and $x$ is the amount of compression past free strain. $U_b$ is the potential energy stored due to bending of the bladder expressed as:

$$U_b = \int_0^L \frac{EI}{2} \kappa^2 dz \,, \tag{6}$$

where $E$ is the Young's modulus of the bladder material, $I = 0.25\pi \left( r_0^4 - r_{inner,0}^4 \right)$ and is the second moment of inertia, and $\kappa$ is the curvature due to the bent shape.

From classical beam theory, the curvature can be approximated as:

$$\kappa = \frac{d^2 y}{dz^2} \,. \tag{7}$$

The transverse deflection $y$ of the bladder in the post-buckling region is given as:

$$y_z = y_{max} \phi_z = y_{max} 2 cos 2\pi L z - 1, \tag{8}$$

which is expressed by multiplying the maximum transverse deflection $y_{max}$, at the middle of the column and the first mode shape of a clamped-clamped column, $\phi(z)$. This first mode shape is used, as higher mode shapes for a column under axial load occur only when the corresponding nodal points are physically restrained [24], which is not true for the case we are analyzing. The bladder is first assumed to have no axial deformation, which allows us to solve for $y_{max}$ by setting the expression for the axial deformation of a largely deformed column to zero [27].

$$\varepsilon_{axis} = \frac{dx}{dz} + \frac{1}{2}\left( \frac{dy}{dz} \right)^2 = 0 \tag{9}$$

$$y_{max} = \frac{\sqrt{4x\left(L_{fs} - x\right)}}{\pi} , \tag{10}$$

where $\varepsilon_{axis}$ is the strain along the axis of the column, which should not be confused with actuator strain. The first term in Equation (9) simply is the strain in the axial direction due to extension. The second term is the contribution of finite rotations in the column to axial strain. The transverse deflection is expressed in terms of the actuator length at free strain $L_{fs}$, which is shown in Figure 3. Substituting the equations for maximum transverse deflection and curvature into Equation (6) and performing the integral gives us an expression of the energy stored in the bladder as a function of the amount compressed.

$$U_b = \frac{4EI\pi^2 x}{\left(L_{fs} - x\right)^2} \tag{11}$$

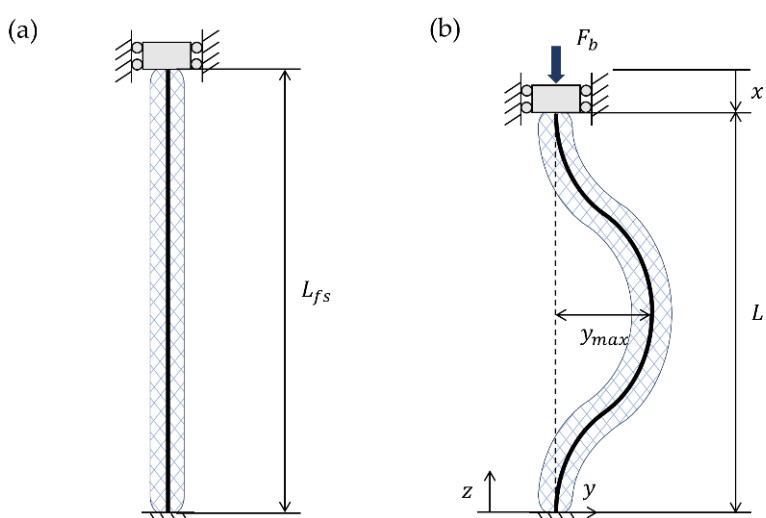

**Figure 3.** Parameters of McKibben actuator at free strain (**a**) and in post-buckling state (**b**).

Performing the first variation of Equation (11) and substituting into Equation (5), the axial force exerted by the bladder due to bending is expressed as:

$$F_b = \frac{4EI\pi^2\left(L_{fs} + x\right)}{\left(L_{fs} - x\right)^3} . \tag{12}$$

It should be noted that the force from Equation (12) is derived under the assumption that no axial deformation occurs. Thus, the arc length of the bladder remains constant and equal to the bladder length at free strain. However, this is not representative of the actual system in which stresses in the axial direction cause some amount of deformation and shortening of the arc length. This change in arc length is not the same as that of a simple column under axial load, as the braid kinematics affect the bladder length as well. Moreover, the actual system is far from ideal, as it contains geometric imperfections, as discussed previously. As a simple approximation of these effects, we propose an equivalent spring system to approximate a corrected bladder force $F_b'$ in the post-buckling region that accounts for axial deformation and non-ideal conditions.

Consider a system of springs that consists of a transverse spring and two springs oriented in the axial direction of the bladder, as shown in Figure 4. The transverse spring constant $k_1$ represents the bending rigidity of the bladder and any compression in the axial direction results in deformation of the transverse spring, causing an opposing axial force.

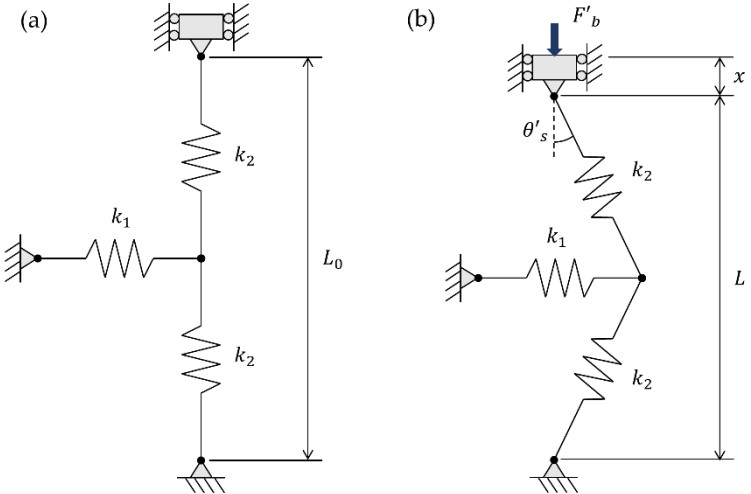

**Figure 4.** Parameters of equivalent spring system in the free strain (**a**) and in post-buckling state (**b**).

The force expression derived in Equation (12) can be considered to be the case in which the axial spring constant $k_2$ is a very large value, such that no deformation occurs in those springs. Calculating $F_b$ and using the corresponding value of $y_{max}$ as the deformation in the transverse spring, an expression for equivalent spring stiffness $k_1$ can be found as:

$$k_1 = \frac{2F_b}{y_{max}sin\theta_s cos\theta_s} , \tag{13}$$

$$\theta_s' = cos^{-1}\left(\frac{L_{fs} - x}{L_{fs} - 2l}\right) , \tag{14}$$

where $\theta_s'$ is the angle between the vertical axis and the axial spring, and $\theta_s$ is a special case for when the deformation in the axial spring $l = 0$ in (14). For a given amount of compression $x$, let us assume this spring stiffness is constant; yet it varies as the actuator is compressed and $x$ changes. Given a value for $k_1$ and assuming a value for $k_2$ as the axial rigidity of the bladder, a relationship can be derived between the deformation in the transverse spring with axial deformation $y_{max}'$ and $l$.

$$y_{max}' = \frac{k_2}{k_1}\frac{4l}{sin\theta_s'} \tag{15}$$

The corrected post-buckling force with axial deformation is expressed as:

$$F_b' = \frac{1}{2}k_1 y_{max}' sin\theta_s' cos\theta_s' = \frac{y_{max}' sin2\theta_s'}{y_{max}sin2\theta_s}F_b . \tag{16}$$

Note that $F_b'$ only accounts for the additional force exerted by the bladder past free strain. The total bladder force is a sum of the bladder force required to reach free strain $F_{bladder}\left(\varepsilon_{free}\right)$, which is the bladder force from Equation (4) evaluated at free strain $\varepsilon_{free}$, and $F_b'$. Furthermore, the total force in the post-buckling region is the sum of forces due to the braided mesh and the bladder, just as it is with the force before free strain. In the typical case of tensile contraction, the mesh deforms axisymmetrically by expanding radially while contracting axially. In such a case, the mesh force is a function of the instantaneous actuator length, as expressed in the ideal mesh force equation. Following the same principle, the mesh force in the post-buckling region remains a function of the braided mesh length; however, that mesh length is no longer the length of the actuator. Since the mesh deforms asymmetrically with some transverse deflection, mesh force becomes a function of the arc length of the buckled shape. The asymmetric mesh force is denoted $F_{mesh}'$, for which computation is identical as Equation (3) but uses the arc length instead of actuator length.

The difference between $F_{mesh}$ and $F'_{mesh}$ is illustrated in Section 3.4. The forces up to free strain for both cases are obviously the same. However, as free strain is reached, the solid lines show axisymmetric deformation, while the dashed lines show the forces as the bladder is bent into its post-buckling shape.

As the bladder is compressed further and continues to bend, the internal moment along the beam increases until it reaches the collapse moment. Due to its complex nature, the collapse moment of elastic inflatable tubes is a standalone research topic and has been extensively studied in various papers [28–31]. The earliest of models assume a cylindrical membrane with an applied internal pressure [26]. These models predict a collapse moment that is a linear function of pressure. Improvements to these models have taken into account the material properties of the bladder and some have used experiments to correct for discrepancies [29,30]. The collapse moment adapted in this paper is a classical formulation in the field of bending inflatable bladders developed by Wood in 1958 [28]:

$$M_{collapse} = \frac{2\sqrt{2}}{9} \pi E r_0 t_0^2 \sqrt{\frac{1}{1-\nu^2} + \frac{4P}{E}\left(\frac{r_0}{t_0}\right)} , \tag{17}$$

where $t_0$ is the initial bladder wall thickness and $\nu$ is Poisson's ratio, which is assumed to be 0.5. Once the maximum internal moment generated in the post-buckled bladder reaches this collapse moment, the model detailed in this section no longer applies and proceeds to the post-collapse region, for which the model is described in the following section.

### 3.3. Post-Collapse Region

The virtual work balance modeling approach taken for the post-collapse region is similar to that taken for the post-buckling region, but with a different mode shape and a torsional spring to represent the "hinge" at the middle of the column due to bladder collapse. The proposed model does not account for the subtle transitional effects that occur between the post-buckling and post-collapse regions, such as the change in cross-sectional area. As a result, the resistive force is expressed in terms of two force equations corresponding to each region, resulting in a piecewise discontinuous model. After the derivation of the post-collapse force model is described, a simple method of making the transition between the two regions will be proposed.

The assumed post-collapse mode shape is shown in Figure 5. It consists of two clamped-free columns connected at their free ends by a torsional spring with constant $k_r$. It is assumed that the torsional spring does not affect the boundary condition of the columns. Upon calculating the internal moment along the column based on the post-buckling mode shape, the magnitude of moment is greatest at each end and at the middle of the column. From the observation of the fact that FAM collapse occurs at one area at which it is most vulnerable, the mode shape is devised to create a hinge at the middle of the column which is represented by a torsion spring with some stiffness. As done for the post-buckling force, the first variation of the total potential energy is set to zero for equilibrium.

$$F_c \delta x = \delta(2U_c + U_r) \tag{18}$$

For the post-collapse region, the strain energy stored in the column consists of energy stored in each clamped-free column and the torsional spring given as:

$$U_c = \int_0^{L/2} \frac{EI}{2} \kappa_c dz \tag{19}$$

and

$$U_r = \frac{1}{2} k_r (2\theta_r)^2 , \tag{20}$$

where $\kappa_c$ is the curvature of the clamped-free column, which is approximated by:

$$\kappa_c = \frac{d^2 y_c}{dz^2} , \tag{21}$$

where the transverse deflection in the post-collapse region $y_c$ is expressed as:

$$y_c = y_{c,max}\phi_c(z) = y_{c,\,max}\left(1 - cos\left(\frac{\pi}{L}z\right)\right),\tag{22}$$

where $\phi_c$ is the first mode shape of the clamped-free column. The torsional spring stiffness $k_r$ is formulated by assuming the bladder has collapsed completely such that the inner walls of the bladder are in contact. Under this assumption, the local radius of curvature is estimated by the wall thickness. The local cross-sectional area is assumed to be an ellipse and the second moment of inertia $I_r = \pi r_0 t_0^3/4$ is used. Therefore, the torsional spring stiffness is expressed as:

$$k_r = \frac{EI_r}{2\theta_r t_0},\tag{23}$$

where $\theta_r$ is the between the column and the horizontal axis, as shown in Figure 5. The angular displacement of the torsional spring is $2\theta_r$ and can be calculated from the slope of the column's free end by taking the derivative of the transverse deflection and evaluating it at the middle. The angle between the column and the horizontal axis is given as:

$$\theta_r = arctan\left(\sqrt{\frac{x}{L_{fs} - x}}\right).\tag{24}$$

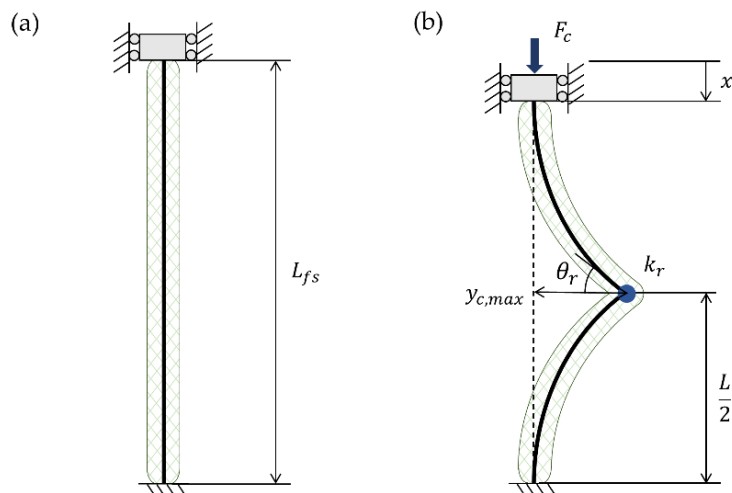

**Figure 5.** Parameters of McKibben actuator at free strain (**a**) and in the post-collapse region represented by the clamped-hinged-clamped column mode shape (**b**).

Taking the first variation of the sum of strain energy terms and substituting into (18) gives the expression for the post-collapse force.

$$F_c = \frac{IE\pi^2 L_{fs}}{4\left(L_{fs} - x\right)^2} + \frac{2k_r\sqrt{x/\left(L_{fs} - x\right)}\,tan^{-1}\left(\sqrt{x/\left(L_{fs} - x\right)}\right)}{x}\tag{25}$$

Combining the post-buckling region force from Equations (16) and (25) results in a piecewise model with a discontinuity at the collapse strain. This is to be expected due to the change in mode shape. The proposed mode shape used for the post-collapse region is representative of the column shape at strains much closer to the maximum strain rather than immediately after the collapse strain. Thus, this mode shape can be used to calculate the force to which the post-collapse region force converges at large strain. In the actual system, a much more gradual transition is observed as the cross-sectional area at the middle of the column deforms from its circular shape to an elliptical shape. The change in cross-sectional area that contributes to the geometric nonlinearity of the system can be applied to both the post-buckling and post-collapse regions. The most widely referenced analysis of this behavior is the study by Brazier on the change in moment due to change

in cross-sectional area shape and the theory on inflatable structures [26,31]. However, we propose a much simpler approach that approximates the transition from the post-buckling to post-collapse force as a first-order response formulated as the following.

$$F'_c = F'_b e^{-\beta(\varepsilon-\varepsilon_c)} + F_c \left(1 - e^{-\beta(\varepsilon-\varepsilon_0)}\right) = F_c + (F'_b - F_c)e^{-\beta(\varepsilon-\varepsilon_c)} \tag{26}$$

The modified post-collapse force begins at the collapse strain $\varepsilon_c$, and is characterized by a transition rate constant $\beta$, which is a parameter that can be tuned manually or through parameter optimization based on empirical data.

*3.4. Summary of Resistive Force Piecewise Model*

The piecewise model for the entire range of strain can summarized as below.

$$F_{total} = \begin{cases} F_{mesh} + F_{bladder} & F_{mesh} > F_{bladder} \\ F'_{mesh} + F_{bladder}\left(\varepsilon_{free}\right) + F'_b & F_{mesh} \leq F_{bladder} \ and \ M_b < M_{collapse} \\ F'_c & M_b \geq M_{collapse} \end{cases} \tag{27}$$

The criterion for the first piece is for strains below free strain. For $F_{mesh}$ and $F_{bladder}$, any of the existing models can be used as long as they can be divided into those two force components. The second and third pieces are for the post-buckling and post-collapse regions, respectively.

Figure 6 shows the FAM force–strain curve for pressures (a) 0 kPa, (b) 34.5 kPa, (c) 137.9 kPa, and (d) 344.7 kPa for a FAM with an initial outer radius of $6.35 \times 10^{-3}$ m (0.5 *in.*), initial bladder wall thickness of $1.6 \times 10^{-3}$ m (0.0625 *in.*), and initial length of 0.127 m (5 *in.*). The initial braid angle was assumed to be 33°. The bladder is a commercial silicone tube with a Young's modulus of 1.78 *MPa* as determined by a simple uniaxial tensile test. Figure 6a shows the resistive force of an inactive FAM. As no pressure is applied, no mesh force is present and only bladder forces exist. Figure 6b shows the resultant total force $F_{total}$, as well as the breakdown of all force components shown in Equation (27). The tensile force region consists of the conventional force–strain relationship up to free strain as shown by the Klute–Hannaford model. After free strain, the post-buckling region begins, and both the mesh and bladder forces deviate from the Klute–Hannaford model. In this region, the bladder is in the post-buckled shape and internal moment is generated along the bladder. When the maximum value of that internal moment exceeds the collapse moment, the resistive force shows a decrease in magnitude (i.e., becomes less negative). This happens during the post-collapse region for which the magnitude of resistive force remains less than its maximum value at collapse. Figure 6b,c illustrates the resistive force regime at higher pressures. For the pressure of 137.9 kPa, only the post-buckling region is shown as the bladder does not collapse before the maximum strain tested. As the applied pressure increases, the free strain increases, thus decreasing the resistive force regime until it is no longer of concern when the applied pressure reaches the maximum operating pressure, as in Figure 6d. Therefore, the significance of the resistive force model is highlighted during lower pressures when the discrepancy between the resistive model and Klute–Hannaford model is greater, as shown by the curves $F_{total}$ and $F_{mesh} + F_{bladder}$ in Figure 6b. As shown in Figure 6a,b, the proposed model shows a good qualitative agreement with the low-pressure FAM behavior past free strain; however, quantitative discrepancies between the model and experimental data exist. To account for such discrepancies, empirical tuning of parameters can be performed in both the tensile and resistive force regimes to yield a better match, as discussed in Section 5.

Figure 7 demonstrates the effect of the transition constant used to characterize the transition from post-buckling to post-collapse region discussed in Equation (26). Lower values of $\beta$s characterize a gradual transition to the post-collapse state. In generating the model curves of Figure 6, a preliminary value of 25 was used. The transition constant is one of the parameters used to empirically tune in Section 5 using an empirically-based parameter tuning.

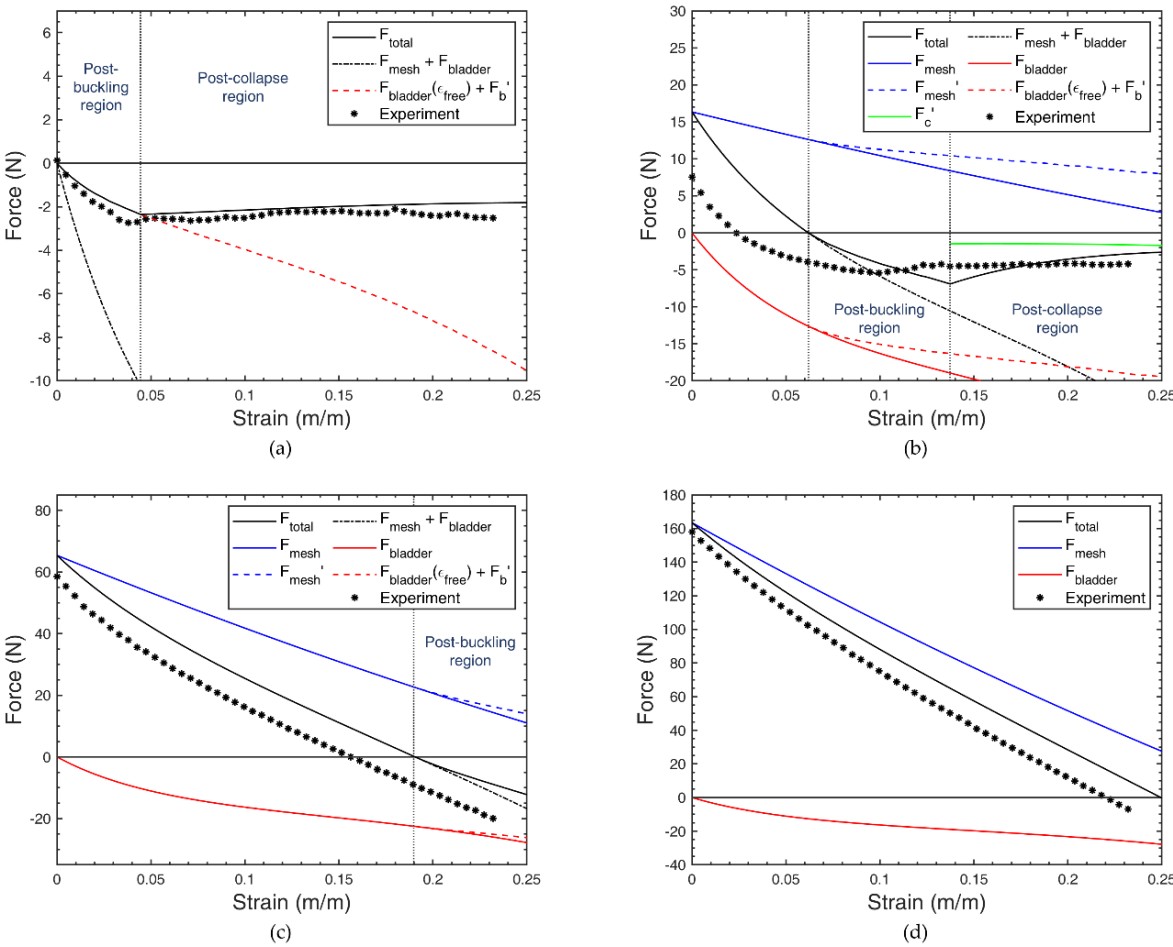

(a)    (b)

(c)    (d)

**Figure 6.** Breakdown of model forces along with corresponding experiment results for (**a**) 0 kPa, (**b**) 34.5 kPa, (**c**) 137.9 kPa, and (**d**) 344.7 kPa. The resultant force from the piecewise model is shown in solid black as well as force components shown in Equation (27).

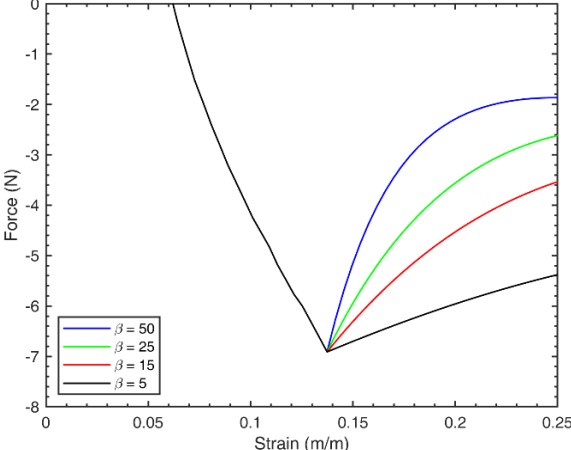

**Figure 7.** Effect of transition constant $\beta$ used to characterize the transition from post-buckling to post-collapse region. The resistive forces are shown for a fluidic artificial muscle (FAM) with identical dimensions as that of Figure 6 and for an applied pressure of 34.5 kPa.

## 4. Effect of Resistive Force on Overall Performance of a Variable Recruitment Bundle

The results of this new model are particularly significant within the context of a variable recruitment bundle. A conventional FAM model, such as the ideal or Klute–Hannaford model, was not intended for forces past free strain and the magnitude of the resistive force

continues to increase as strain increases past free strain. During operation of a variable recruitment bundle, inactive or low-pressure FAMs are compressed significantly past their free strain, for which conventional models would predict a much greater magnitude of resistive force than what is present. Instead of a steady increase in resistive force magnitude past free strain, as shown by $F_{mesh} + F_{bladder}$ in Figure 6, the proposed model better predicts the force behavior of FAMs past free strain. Therefore, this model is more suited to predict the overall force output of a variable recruitment bundle.

The force–strain curves for a simple variable recruitment bundle actuator consisting of two MUs, each made of one identical FAM, were generated. The bundle can be in two different states of operation called recruitment states (RSs). The two primary variable recruitment schemes that have been used in previous work are called batch recruitment and orderly recruitment [10,32]. Orderly recruitment will be assumed for this analysis, and Figure 8 shows a graphical depiction of the activation sequence for this recruitment scheme. In the first recruitment state, only one MU is activated (i.e., pressurized), and the other MU is inactive. A recruitment state can either be partially or fully activated depending on the pressure supplied. It is fully activated when the corresponding MU pressure is at its maximum.

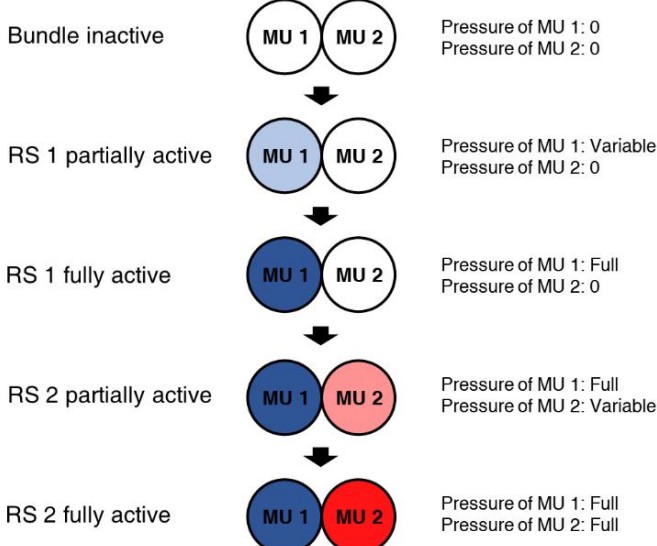

**Figure 8.** Orderly recruitment activation scheme for a bundle with two recruitment states (RS), denoted as RS 1 and RS 2. The bundle is in RS 1 when only the first motor unit (MU) is active. The bundle is in RS 2 when both MU 1 and MU 2 are active.

### 4.1. Quasi-Static Force–Strain Space for Variable Recruitment Bundle with Resistive Forces

Figure 9 shows the force–strain curves for a variable recruitment bundle with two recruitment states with and without resistive forces included in the analysis The force curves plotted in blue show the force–strain relationship of the first recruitment state. Once the first recruitment state is fully activated, the bundle can enter its second recruitment state, during which the first MU stays at its maximum pressure and a second MU is either partially or fully activated. The red isobaric curves show the force–strain space for the second recruitment state. To illustrate the increase in pressure of the MU of a recruitment state, higher pressures are depicted in relatively thicker lines. In this scenario, quasi-static force curves were plotted for pressures between 0 and 413.7 kPa at 68.9 kPa intervals. The FAMs used in this simulation are assumed to have an initial length of 0.127 m, initial outer radius of 0.009 m, a radius-to-wall thickness ratio of 0.25, and an initial braid angle of 33°. The force–strain space of Figure 9a is generated by neglecting the resistive forces and assuming any forces past free strain to be zero. A more realistic case is presented in Figure 9b, in which the resistive forces are included in the overall bundle force.

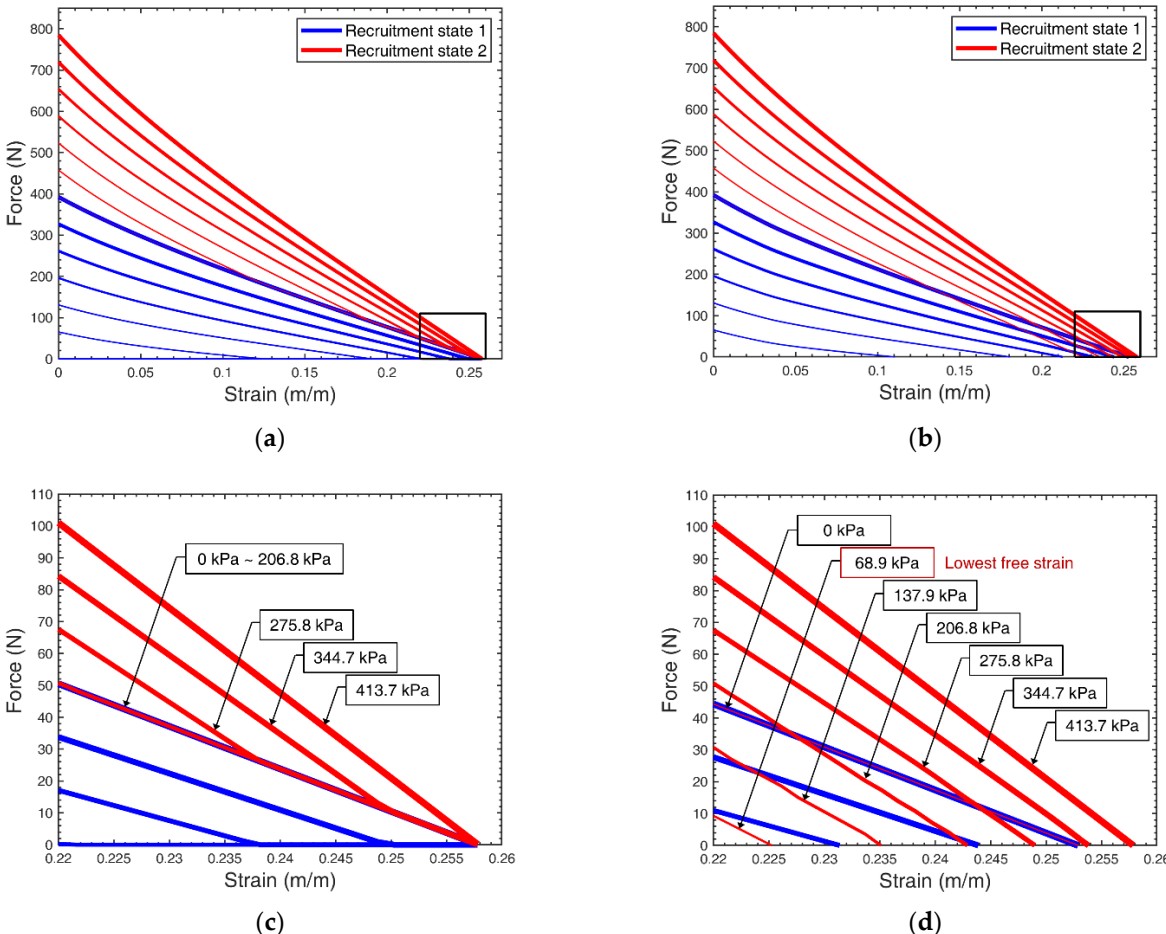

**Figure 9.** Force–strain space of a variable recruitment bundle with two motor units (MUs) of equivalent cross-sectional area with (**a**) resistive forces neglected, and (**b**) resistive forces considered. (**c**,**d**) show zoomed-in views of the force–strain space near maximum strain with resistive forces neglected and included, respectively. Increasing pressure values are represented by increasing thickness of lines. In recruitment state 2 (red lines), the pressure applied to MU 1 is constant at 413.7 kPa, while the pressure applied to MU 2 is indicated in (**c**,**d**) to show the progression from inactive to fully active.

The key difference observed from the force–strain space of a bundle with resistive effects is an overlapping region between the first and second recruitment states that becomes prominent during strains near the maximum bundle free strain. As shown in Figure 9a, the boundary between recruitment states 1 and 2 is distinct when resistive forces are neglected. As the first MU is fully activated, any additional pressure supplied to the second MU, transitioning to the second recruitment state, either increases or maintains the force output of the bundle. In Figure 9c, the pressures applied to MU 2 are indicated for a bundle model with resistive forces are neglected. When a pressure of 275.8 kPa is supplied to the second MU, the model predicts that the force will increase for strain values less than 0.24. However, applying the same pressure for strains greater than 0.24 is predicted to not affect the force output of the bundle. This occurs because free strain is pressure-dependent and the free strain generated by the MU 2 FAM at these pressures is less than the bundle strain. Thus, for a specific force–strain output, the required operating state of the bundle can be easily categorized into either recruitment state 1 or 2. However, when resistive forces are considered, the negative forces due to resistive effects cause the forces in recruitment state 2 to actually fall below that of recruitment state 1 for certain strains. Therefore producing a region of overlap between the domains of each recruitment state. Within this overlapping region, the bundle can output the same force and strain by operating in either recruitment state. In fact, as recruitment state 2 is activated and the pressure in MU 2 is increased, the isobaric force curve of recruitment state 2 recedes back

into the force–strain space of recruitment state 1, as shown in Figure 9d. The effect of this phenomenon is discussed in further detail in Section 6.

*4.2. Efficiency Analysis for Isobaric and Isotonic Contraction*

One of the advantages of variable recruitment lies in its increased overall efficiency. Therefore, it is worth exploring the efficiencies of variable recruitment bundles when resistive forces are considered. By choosing a recruitment literate state that minimizes the energy input while meeting the force requirements, variable recruitment has the potential to increase overall efficiency when compared to a single equivalent motor unit (SEMU). The efficiency of an actuation reaching a given point in the force–strain space from zero strain is the ratio between mechanical work output and fluid energy input where the expression for mechanical work output is given as:

$$W_{mech,\ out} = \int_0^{\varepsilon L_0} F_{bundle} dx \ , \tag{28}$$

where $F_{bundle}$ is the sum of all the forces generated by individual FAMs calculated by the proposed model, including the negative forces caused by resistive effects. The fluid energy input is defined as:

$$E_{fluid,\ in} = P_s \Delta V \ , \tag{29}$$

where $P_s$ is the source pressure applied to the valve before losses due to throttling occur. The change in volume to reach a specific strain is calculated by subtracting the fluid volume at that strain value from the initial fluid volume. The fluid volume at a specific strain value is calculated by solving for inner radius $r_{inner}$ as a function of strain by assuming the bladder material is incompressible. Finally, the expression for efficiency is as follows:

$$\eta = \frac{W_{mech,\ out}}{E_{fluid,\ in}} \ . \tag{30}$$

To highlight the system efficiency over the entire force–strain space, two simple cases of quasistatic contraction are considered: isobaric contraction and isotonic contraction. During isobaric contraction, the pressure supplied to the FAM is kept constant while it contracts with varying force. For isotonic contraction, the load is kept constant while the FAM contracts under varying pressure. This is equivalent to moving horizontally through the force–strain space. For each point in the force–strain space, the efficiency for a FAM to move from free length to the strain at the specific point is calculated. The resulting isobaric and isotonic efficiencies for a bundle with two recruitment states are plotted in Figure 10a,b. As discussed in Section 4.2, an overlapping region exists for an actuator bundle when resistive forces are considered. The efficiency plots in Figure 10 show only the higher efficiency value within that region.

To study the potential detrimental impact of resistive forces on the overall efficiency of a variable recruitment bundle, the average efficiencies with resistive effects considered and neglected are compared in Figure 10c,d. Prior literature using simpler models that neglect resistive forces predicted that a variable recruitment bundle would show increased efficiency compared to a SEMU with identical total cross-sectional area [9,32]. As the number of MUs (and therefore recruitment states) is increased, the efficiency continues to increase, but at a decreasing rate. This trend still holds when resistive forces are considered, producing only a minor decrease in efficiency, as shown in Figure 10c,d. As its name suggests, resistive forces act against the tensile force output of active MUs and lower the mechanical work output. However, if the number of recruitment states is increased while keeping the overall bundle cross-sectional area constant, the slenderness ratio of each MU, defined as $L_0/r_0$, increases. As a result, the resistive force of each MU decreases, along with the impact it has on the overall efficiency of a bundle. Consider a comparison between a bundle with two MUs (as simulated in this study) to a bundle with five MUs with the same overall bundle cross-sectional area. Assume each bundle is operating in its first recruitment state. For the two-MU bundle, the resistive forces generated by the single inactive MU are greater than the resistive forces generated by the four inactive MUs in the bundle with

five MUs. This is because to keep the cross-sectional area of both bundles the same, the bundle with five MUs must contain FAMs that are much slenderer. The resistive force model in this paper predicts that resistive forces decrease with slenderness ratio, which is why we see very little decrease in efficiency, even as the number of MUs within a bundle is increased. Although not within the context of a variable recruitment bundle, Suzumori et al. investigated the mechanics of a bundle of multiple thin McKibben actuators, showing a decrease in bending rigidity of the overall bundle compared to a single large diameter FAM [33]. Therefore, while resistive forces affect the overall force–strain space, which has implications on the control and switching of recruitment states (to be discussed further in the following sections), their overall detriment to the efficiency of the bundle is negligible.

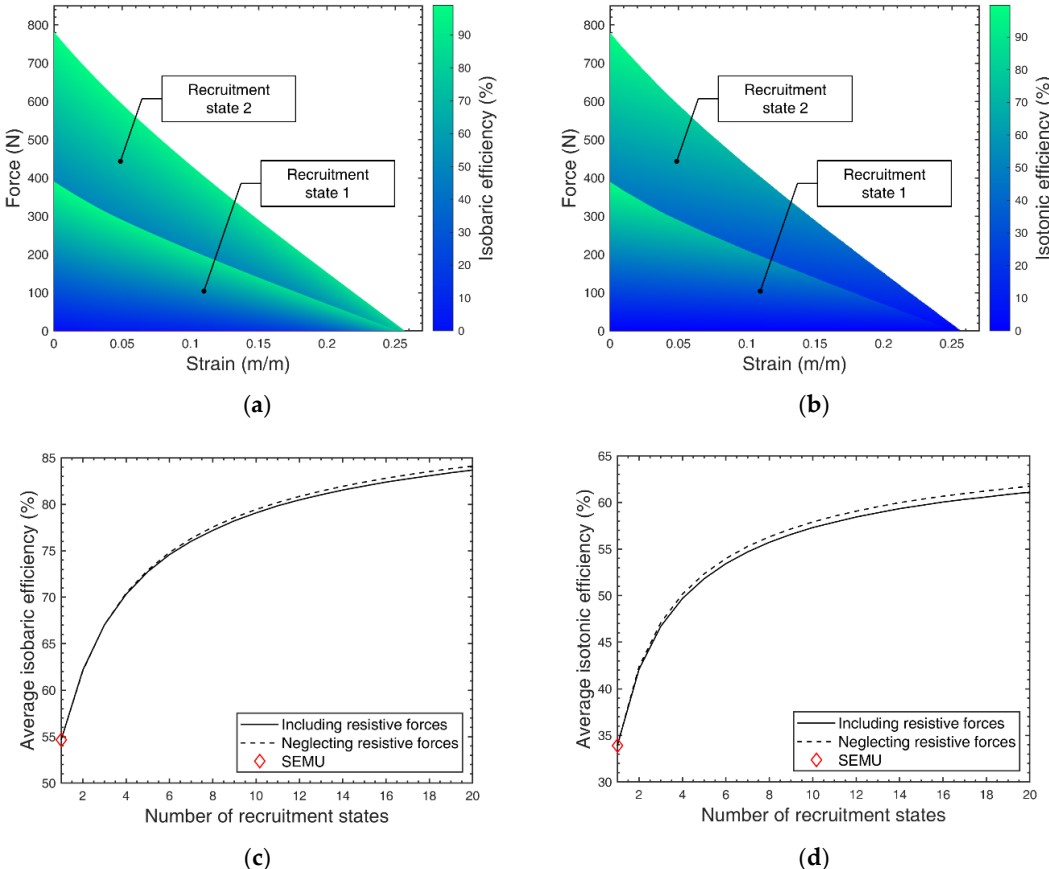

**Figure 10.** The efficiency of the bundle to arrive at a point in the force–strain space for (**a**) isobaric and (**b**) isotonic contractions. For isobaric contraction, the bundle is operated along the constant pressure curves, thus the force varies while contracting. For isotonic contraction, the load is held constant while the bundle contracts, thus moving horizontally within the force–strain space by varying pressure. The average isotonic and isobaric efficiencies are plotted versus number of recruitment states and shown in (**c**,**d**), respectively. The efficiency for a single equivalent motor unit (SEMU) is indicated as well.

## 5. Improving the Model Through Empirical Parameter Tuning

While the model derived above provides a predictive and qualitatively reasonable analysis for understanding the implications of resistive forces on variable recruitment bundles, the model can be empirically corrected to improve quantitative agreement with experimental FAM characterization data. Such semi-empirical modeling has been shown to useful for model-based feedforward control of artificial muscles [34], and in simulation tools for understanding the implications of different recruitment strategies [10] or hydraulic system topologies [15] on FAM actuation systems. Therefore, to further improve the model to match the force from experiments, a method of force correction was developed. There are models in the literature that account for advanced effects such as the tapering effect at the ends of the FAM and the hyperelasticity of the bladder on the tensile force generation

of the FAM [19,21–23]. However, it is often the case that the blocked force and free strain is overpredicted even when calculated with these advanced models, due to uncertainty in the material properties or imperfections in the components used to build the bladder. Tondu and Lopez used a tuning parameter that can either be a constant or a function of pressure to modify the ideal force model shown in Equation (3) to account for the pressure-dependent free strain [19]. Building upon this method, Meller et al. used an additional parameter that was acquired from experiment data to correct for the discrepancy in blocked force [14]. These methods were used to modify the ideal force model, which does not consider forces due to the bladder. Similar to the method proposed by Meller, we employed two tuning parameters, based on the blocked force and free strain information from experimental data, to modify both the mesh force (ideal force) and bladder force in the tensile region. After these two parameters were used to correct for the tensile region in the piecewise model, additional parameters for the resistive forces were optimized using empirical data.

### 5.1. Tensile Force Correction

The modified model for forces up to free strain is expressed as:

$$F_{mod} = C_2(F_{total} - C_1) , \tag{31}$$

$$F_{mesh, mod} = C_2 F_{mesh} , \tag{32}$$

and

$$F_{bladder, mod} = F_{mod} - F_{mesh, mod} , \tag{33}$$

where the forces prior to modification are equivalent to those from Equation (4) and the strain tuning parameter $C_1$ is found by evaluating the theoretical force at the free strain obtained from experiment, known as $\varepsilon_{free,exp}$, in the equation:

$$C_1(P) = F|_{\varepsilon_{free, exp}} . \tag{34}$$

In addition, the force tuning parameter $C_2$ is expressed in terms of the experimental block force $F_{blocked, exp}$, theoretical block force $F_{blocked}$, and $C_1$ as:

$$C_2(P) = \frac{F_{blocked, exp}}{F_{blocked} - C_1} . \tag{35}$$

Both tuning parameters are functions of pressure that are specific to an actuator. Note that the unit of $C_1$ is in Newtons and $C_2$ is unitless. Both factors are greater than zero and have no upper bound. The parameters were obtained by measuring the free strain and blocked force as functions of pressure, much like the procedure followed by Meller et al. [14]. As shown in [14], the function for free strain typically followed a low-order polynomial and blocked force typically increased linearly with pressure with a positive force x-intercept.

### 5.2. Resistive Force Correction

Once the forces before free strain were corrected based on empirical data, several parameters of the proposed model for the post-buckling and post-collapse regions were optimized based on experimental data. A least-squares fit analysis was conducted to determine correction factors for the following parameters.

1. Young's Modulus, $E$
2. Collapse moment, $M_{collapse}$
3. Transition constant, $\beta$
4. Torsional spring stiffness, $k_r$

The resistive force model does not consider the hyperelasticity of the bladder as the FAM is compressed. The bladder used for the experiments was an off-the-shelf component without any specification of material properties. To account for such uncertainties, a correction factor was used to optimize the Young's modulus and the optimized value was used across all pressures.

Furthermore, a correction factor was optimized to modify the collapse moment. Wielsgosz et al. used a factor of $0.25\pi$ to modify the collapse moment as experimental validation

showed a tendency to overpredict the value [35]. Stephans et al. investigated the correlation of collapse moment to the slenderness ratio of the pressurized cylinder. For cylinders with low slenderness ratios, the collapse moment tends to be closer to the "classical" formulation, whereas for that of higher slenderness ratios, the collapse moment reaches the load formulated by Brazier [36]. A similar correction factor is used to modify the collapse moment given in Equation (17).

The third tuning parameter is the transition rate constant $\beta$ given in Equation (26), used in characterizing the transition from post-buckling to post-collapse regions. Lastly, a correction factor is used to optimize the torsional spring stiffness $k_r$ used in Equation (20) to model the post-collapse "hinge" at the middle of the bladder.

### 5.3. Experiments to Generate Correction Factors

Experiments were performed on an in-house built linear dynamometer developed by Chipka et al. [37], as shown in Figure 11. The air pressure applied to the FAMs was controlled in closed-loop by pneumatic servo valves (FESTO MPYE-5-M5-010-B), while the drive cylinder is hydraulically powered and controlled by a MOOG Series G761-3005B servo valve. A Hydraulic power unit (HPU, Haldex GC9500) provided power to a drive cylinder that was used to control the contraction of the FAM. The FAM was constrained between two plates with a load cell (Transducer Techniques SSM-1K) to measure the FAM axial force. Prior to attaching the FAM to both plates, it was pressurized at the maximum test pressure and allowed to contract freely, and its maximum strain was measured. This value was used to determine the stroke of the drive cylinder. The FAM was then attached to both plates separated by the amount of actuator free length which is determined by adjusting the starting position of one of the plates until the axial force measured was zero. From this starting position, the FAM was pressurized to the maximum testing pressure and repeatedly contracted several times prior to collecting force data to account for Mullins effect [38]. Force data was measured for pressures from 0 to 413.7 kPa (60 psi) in 34.5 kPa (5 psi) intervals. The position of the drive cylinder was controlled at a sufficiently slow rate to remove any dynamic effects and yield a quasi-static measurement. The force for two FAMs of different slenderness ratios were measured to demonstrate its correlation to resistive force magnitude and support the analytical efficiency hypothesis stated in Section 4.2. An initial radius of 0.0635 m, initial braid angle of 33° was used for both FAMs with an initial length of 0.102 and 0.127 m, corresponding to slenderness ratios of 8 and 10, respectively.

### 5.4. Results from Empirical Parameter Tuning

The correction factors, which are a result of least-squares optimization, are summarized in Table 1. The correction factors for the Young's modulus, collapse moment, and transition constant were the same for all pressures. The correction factor for the torsional spring constant in the post-collapse region showed a dependence on pressure since the torsional spring constant calculation proposed does not consider the increase in bending rigidity due to increased pressure.

The measured force data from experiments are shown in Figure 12 along with the simulated force curves from the empirically corrected model. The force curves from 0 to 137.9 kPa in 34.5 kPa intervals are shown as resistive force, bearing more significance in low pressure ranges. In the resistive force regime, a good agreement is observed between the experiments and the corrected model. Free strains for constant pressure force curves past 137.9 kPa occur closer to the maximum free strain and do not show the collapse of the bladder. In the tensile force regime, the model blocked force and free strain obviously match those of the experiment as a result of the empirical tuning described in Section 5.2. Although the cross-sectional area is the same for both FAMs, a difference in blocked force is observed due to a difference in bladder force at zero strain. A significant difference in resistive force magnitude is observed between the two FAMs with different slenderness ratios. As predicted in the model, the FAM with a lower slenderness ratio exhibited a larger magnitude of resistive force. This supports the conclusion that efficiency does not decrease as the number of MUs increases due to slenderness ratio effects, as detailed in Section 4.2.

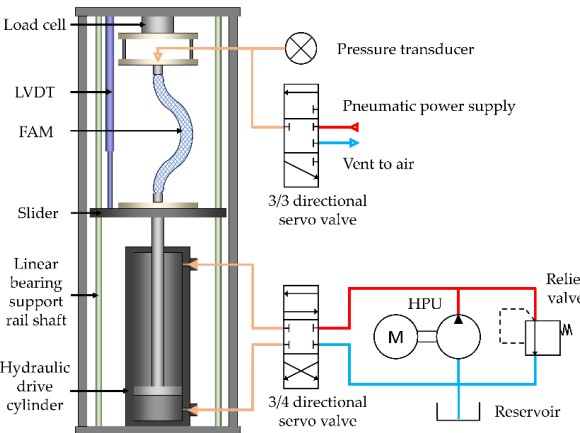

**Figure 11.** Experiment setup using the linear hydraulic dynamometer (LHD) developed by Chipka et al. [37]. A hydraulic power unit (HPU) is used to actuate the drive cylinder. The fluidic artificial muscle (FAM) is activated using a pneumatic power supply while the force and contraction is measured using a load cell and a linear variable differential transformer (LVDT), respectively.

**Table 1.** Resulting correction factors from empirical parameter tuning.

| FAM No. | Slenderness Ratio, $L_0/r_0$ | Correction Factors | | | Transition Constant, $\beta$ |
| | | Young's Modulus | Collapse Moment | Torsional Spring Constant | |
| --- | --- | --- | --- | --- | --- |
| 1 | 8 | 1.25 | 1.1 | $0.87P + 1.25$ | 100 |
| 2 | 10 | 0.95 | 0.75 | $0.92P + 0.83$ | 100 |

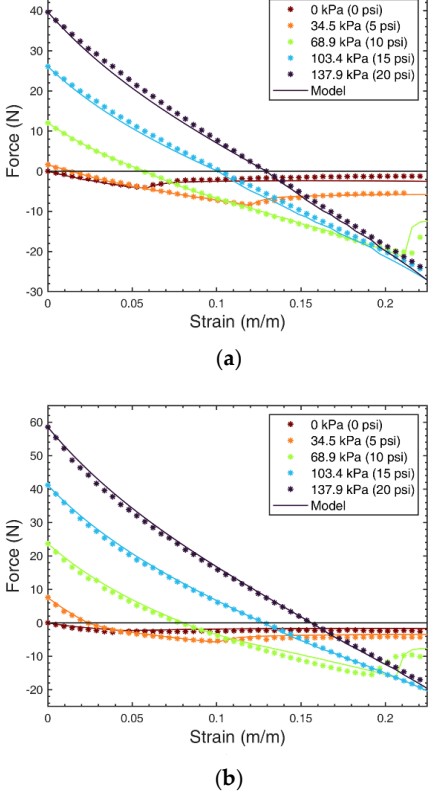

**(a)**

**(b)**

**Figure 12.** Result of empirical-based tuning of parameters and comparison to experimental data for a FAM with (**a**) slenderness ratio of 8 and (**b**) slenderness ratio of 10. Forces for 0, 34.5, 68.9, 103.4, and 137.9 kPa are measured and used to optimize the parameters specified in Table 1. The Young's modulus, collapse moment, and transition rate constant are independent of pressure, while the torsional spring stiffness varies with pressure.

## 6. Bundle Free Strain Gradient Reversal

The force–strain plots shown in Section 4 of this paper illustrate an interesting phenomenon that occurs within a bundle when resistive forces are considered. When MU 1 reaches source pressure, MU 2 is activated and the bundle transitions between recruitment states 1 and 2. Once MU 2 is activated, the overall bundle free strain actually begins decreasing because resistive forces are stronger in low-pressure FAMs than in inactive FAMs. As MU 2 pressure is further increased, the free strain continues to decrease, until eventually, the kinematic forces generated by the FAM in MU 2 overcome the resistive forces, and free strain begins to increase. We call this phenomenon free strain gradient reversal, and the point at which the free strain ceases to decrease due to the presence of MU 2 is called the free strain gradient reversal point. This idea can be more clearly illustrated in Figure 13, which is a plot of overall bundle free strain vs. MU 2 pressure for different models: the ideal model, the Klute–Hannaford model [20], the corrected Klute–Hannaford model neglecting resistive forces discussed in Section 5.1, and the corrected Klute–Hannaford model including resistive forces, as well as a plot for a two-MU experiment. The experiment results shown in Figure 13 are measurements obtained from the FAM with a slenderness ratio of 10 in Section 5.4.

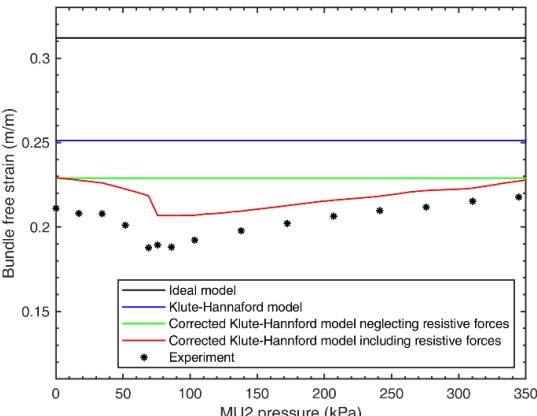

**Figure 13.** Free strain of actuator bundle versus pressure applied to motor unit (MU) 2 shown for the ideal model, the Klute–Hannaford model [20], the corrected Klute–Hannaford model neglecting resistive force, the Klute–Hannaford model including resistive forces, and experimental data. The pressure of MU 1 is kept constant at 344.7 kPa. The free strain initially decreases in response to increased pressure in the second recruitment state.

From Figure 13, we can see that previous models did not capture the gradient reversal phenomenon that exists due to the presence of resistive forces, but the model presented in this paper demonstrates this behavior. Every data point from the experiments falls within 10% of the predicted values, with the exception of one, which falls within 15%. The differences between the model and the experiment are believed to be largely due to fabrication uncertainties in the FAMs, which were constructed to be identical, but may have had slightly different free lengths or initial braid angles. Such fabrication discrepancies result in slightly different blocked force and free strain for the two FAMs, leading to a different bundle blocked force and free strain than that predicted by the model. We can investigate this gradient reversal further by considering the purely analytical case of an actuator bundle consisting of five FAMs. We compare two different bundle configurations: one bundle with two MUs (MU 1 having one FAM and MU 2 having four FAMs) and another bundle with five MUs (one FAM per MU). The comparison plot of these two bundle configurations is shown in Figure 14. The experimental results for a FAM with slenderness ratio 10 from Section 5.4 are used to generate these plots. Similar to the two-MU bundle in Figure 13, the pressure of MU 1 is kept at 344.7 kPa.

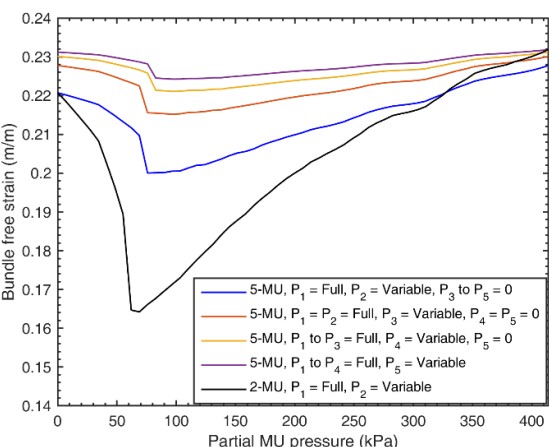

**Figure 14.** Model comparison of bundle free strain vs. the motor unit (MU) pressure while transitioning between recruitment states 1 and 2 for a bundle consisting of five FAMs. $P_n$ denotes the pressure applied to MU $n$. The black line represents a bundle with two MUs, one containing a single fluidic artificial muscle (FAM) and the other containing four FAMs. The colored lines represent the different recruitment states a of a five-FAM bundle that consists of five MUs (i.e., one FAM per MU).

The results from Figure 14 show that, for the two-MU bundle configuration, the reduction in free strain due to the resistive forces during the transition from recruitment state 1 to recruitment state 2 is much more pronounced than the reduction in free strain associated with transitioning from recruitment states 1 through 5 for the five-MU bundle. This result is particularly significant when considering bundle design, because we can change the characteristics of a bundle without changing the number of FAMs in the bundle simply by changing the distribution of MUs within the bundle. In future work, we will investigate optimal methods to distribute these MUs to accomplish specific actuation tasks.

## 7. Conclusions

In this paper, the reaction forces of FAMs compressed axially past free strain, defined in this paper as resistive forces, have been modeled. Furthermore, the effect of resistive forces on the overall performance of a variable recruitment bundle was brought to attention. The resistive force of a FAM is divided into two regions: post-buckling and post-collapse. In the post-buckling region, the additional force required to bend the bladder into the post-buckling shape was derived using a virtual work balance. To account for geometric imperfections and axial deformation of the bladder, an equivalent spring system was proposed to refine the force in the post-buckling region. The termination of the post-buckling region is determined by when the internal moment generated in the bladder exceeds the collapse moment. After the bladder collapses, a similar approach is taken for the post-collapse region, in which a clamped-hinged-clamped mode shape is used to calculate the force required to bend the bladder into that shape. A first-order response is used in the post-collapse region to approximately capture the transition from post-buckling to post-collapse behavior.

To further improve the model to better match experimental results, a technique that uses two pressure-dependent tuning parameters to correct for the discrepancy in blocked force and free strain was proposed for forces up to free strain. Unlike other methods in the literature, the proposed method can modify the mesh and bladder force components. After tuning the parameters for forces up to free strain, several parameters of the proposed resistive force model are optimized using empirical data. The Young's modulus, collapse moment, and transition rate constant were determined to be constant for a given FAM geometry and independent of pressure. Only the torsional spring stiffness used in the post-collapse model was shown to be pressure-dependent. As a result, a viable semi-empirical physics-based model was validated with experimental results.

The force–strain space for a bundle with two recruitment was simulated to show an overlapping region between recruitment state 1 and 2, which was not observed until resistive forces were considered. An efficiency comparison between bundles with and without resistive forces showed a negligible amount of difference in both isobaric and isotonic efficiency, indicating that the presence of resistive forces in FAM bundles do not preclude variable recruitment from be used as energy-saving strategy. As the number of recruitment states increased, the slenderness ratio of FAMs increased, resulting in a decrease in resistive force magnitude. The addition of the resistive force exerted by the inactive/low-pressure MUs to the force output of active MUs results in a strain-dependent change in performance when advancing to a higher recruitment state. This free strain gradient reversal is significant when considering the problem of how to optimize bundle design how to develop control criteria for switching in between recruitment states, which is to be investigated in future work.

The post-buckling load-deflection behavior of an inflatable tube under axial load is a topic of extensive research and not a simple problem when applied to FAMs. However, a relatively simple solution can be derived using the assumptions in this paper that will allow us to better evaluate bundle performance and design more effective variable recruitment controllers.

**Author Contributions:** Conceptualization, J.Y.K., N.M. and M.B.; funding acquisition, M.B.; investigation, J.Y.K. and N.M.; methodology, J.Y.K. and N.M.; resources, M.B.; supervision, M.B.; Validation, J.Y.K.; visualization, J.Y.K. and N.M.; writing—original draft, J.Y.K. and N.M.; writing—review and editing, M.B. All authors have read and agreed to the published version of the manuscript.

**Funding:** This work was supported primarily by the Faculty Early Career Development Program (CAREER) of the National Science Foundation under NSF Award Number 1845203 and Program Manager Irina Dolinskaya. Additionally, this material is based upon work supported by the National Science Foundation Graduate Research Fellowship Program under Grant No. 1650114. Any opinions, findings and conclusions or recommendations expressed in this material are those of the author(s) and do not necessarily reflect those of the National Science Foundation.

**Institutional Review Board Statement:** Not applicable.

**Informed Consent Statement:** Not applicable.

**Data Availability Statement:** Data is contained within the article.

**Conflicts of Interest:** The authors declare no conflict of interest.

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
