# Peer review of "Modeling of Resistive Forces and Buckling Behavior in Variable Recruitment Fluidic Artificial Muscle Bundles"

_actuators, doi:10.3390/act10030042_

Round 1

Reviewer 1 Report

The paper presents systematic research of the reaction forces of FAMs compressed axially past free strain. The modeling of resistive force for the post-buckling region and post-collapse region is built up, respectively. The implications of resistive forces on the overall performance of a variable recruitment bundle are elaborated properly and the new model is proved to be more suited to predict the overall force output compared with conventional FAM models. Further, an empirically-based correction method is proposed in order to better match experimental results. Interestingly, a phenomenon called “free Strain Gradient Reversal” is introduced at the end of the manuscript because the resistive forces are taken into account. Overall, this paper is a topic of interest to the researchers in the related areas and well-written but needs slight improvement before acceptance for publication.

Detailed comments are as follows:

  1. Ref.9 is not cited in the text, I guess it is supposed to be cited in line 45. [Page 2, Line 45]
  2. The first mode shape of buckling of an axially compressed fixed-fixed bar is according to Ref. 25, which is different from Eq.8 in the manuscript. Obviously, y(z)=0 when z=0 or L, which is inconsistent with Eq.8 in the manuscript. [Page 6, Line 190-191]
  3. Similar question for Eq.22, please check whether the deflection yc is expressed correctly. [Page 9, Line 282-283]
  4. Please check whether the citation method of Reference 28 is accurate, it is a bit difficult to find the source of the citation. [Page 6, Line 196]
  5. Except for the midpoint, will the two ends of the bladder also deforms from their initial circular shape when the bladder enters the post-collapse region? Just as the MU on the far right in Fig.1 and if so, will it affect the modeling results? [Section 3.2]
  6. The pressure value corresponding to each colored curve should be illustrated in Fig.11 or stated in the figure caption for clarity. [Page 18, Line 560]

Reviewer 2 Report

Generally, this is an outstanding presentation and should be published with minor revisions. My comments are below:

  1. I recommend not using punctuation in equations (especially end commas and periods.) Not necessary and can be confusing, and use in not entirely consistent in paper.
  2. Eq. 26: Align equal signs
  3. Eq. 27: use of commas is very odd... suggest removal
  4. section 5.3: It would be impossible from the information given to replicate this work. A figure or block diagram and a better explanation of the set up is needed.
  5. The authors are concerned with buckling of muscles that have essentially zero inflation pressure. Could this be mitigated somewhat by leaving a small pressure in the muscle to avoid premature buckling or is this not feasible? If this is a problem are there mitigation strategies or is this not an issue?

Reviewer 3 Report

This paper developed a model for resistive forces and studied their implications on the overall force output of a variable recruitment bundle. This method is relatively simple and allow researchers to better evaluate bundle performance and design more effective variable recruitment controllers. The research topic is quite interesting and the overall presentation of this paper is smooth.
